# A Low-Latency and Energy-Efficient Neighbor Discovery Algorithm for Wireless Sensor Networks [note 1]

**DOI:** 10.3390/s20030657

**Published:** 2020-01-24

**Authors:** Zhaoquan Gu, Zhen Cao, Zhihong Tian, Yuexuan Wang, Xiaojiang Du, Guizani Mohsen

**Affiliations:** 1Cyberspace Institute of Advanced Technology, Guangzhou University, Guangzhou 510006, China; zqgu@gzhu.edu.cn; 2Department of Computer Science, Rice University, Houston, TX 77025, USA; zhen.cao@rice.edu; 3College of Computer Science and Technology, Zhejiang University, Hangzhou 310027, China; 4Department of Computer and Information Sciences, Temple University, Philadelphia, PA 19122, USA; dxj@ieee.org; 5Computer Science and Engineering Department, Qatar University, Doha 2713, Qatar; mguizani@ieee.org

**Keywords:** neighbor discovery, wireless sensor networks, communication collision, discovery latency, duty cycle

## Abstract

Wireless sensor networks have been widely adopted, and neighbor discovery is an essential step to construct the networks. Most existing studies on neighbor discovery are designed on the assumption that either all nodes are fully connected or only two nodes compose the network. However, networks are partially connected in reality: some nodes are within radio range of each other, while others are not. Low latency and energy efficiency are two common goals, which become even more challenging to achieve at the same time in partially connected networks. We find that the collision caused by simultaneous transmissions is the main obstruction of achieving the two goals. In this paper, we present an efficient algorithm called *Panacea* to address these challenges by alleviating collisions. To begin with, we design *Panacea*-NCD (*Panacea* no collision detection) for nodes that do not have a collision detection mechanism. When *n* is large, we show the discovery latency is bounded by O(n·lnn) for any duty cycle (the percentage time to turn on the radio), where each node has *n* neighbors on average. For nodes that can detect collisions, we then present *Panacea*-WCD which also bounds the latency within O(n·lnn) slots. Finally, we conduct extensive evaluations and the results also corroborate our analyses.

## 1. Introduction

Wireless sensor networks have been widely adopted in a wide range of Internet-of-Things (IoT) applications such as smart cities [1], health-care systems [2,3], intelligent detection [4,5], and remote monitoring [6,7,8]. In general, a large number of sensor nodes would be dispatched or deployed in some monitoring area to collect or aggregate information for further intelligent decisions. With the fast development of sensor system and communication technology, these sensor nodes could be deployed in a distributed manner without any pre-defined network topology; they can compose a wireless network to exchange information with the help of advanced communication technologies, such as bluetooth, wifi, 5G, etc.

*Neighbor discovery* is a fundamental process of constructing the wireless network among the sensor nodes, where each sensor node could discover other sensors within its communication range. As a cornerstone and an essential step in configuring wireless sensor networks, neighbor discovery has been extensively studied in the past few years [9,10,11,12,13,14,15,16,17,18,19,20,21,22] and the main objective of these algorithms is to reduce the discovery latency for a specific node or the whole network.

The extant algorithms can be classified into two categories. The probabilistic methods exploit the probability theory to reduce the communication collision caused by multiple transmissions simultaneously. These methods could make sensor nodes discover the neighbors within an expected low latency, but they all assume the network is fully connected where any two nodes can communicate directly. However, a large number of sensor nodes would be deployed in a large area and these nodes cannot communicate directly when their distance is far away or some nodes are restricted due to extreme environment. The deterministic methods could guarantee the discovery process where the sensor nodes only turn on their radios for a short time, but these methods are applicable to two nodes since they do not consider the collisions that may exist among a large number of sensors. Hence, these methods cannot be adopted directly for a large number of sensors, as the collisions would affect the normal communication process. In designing efficient neighbor discovery algorithms for wireless sensor networks in IoT applications, there are two main challenges. First of all, sensor nodes have limited battery since the size of the sensors is small and battery recharging would be costly in applications like agricultural monitoring or underwater detecting; the node could turn on its radio only for sufficient communication and keep the radio off to save energy. Second, some IoT applications need a large number of sensors that are distributed in a large area or in some hostile environment; the sensors can only compose a partially connected network in which some nodes are not within the radio range or some nodes can be detected with some probability due to the obstacle or the block in the environment. However, the extant works cannot solve the two challenges concurrently.

To save energy, sensor nodes switch their radios off for most of the time and only turn the radios on for necessary communication. The fraction of time a node turns its radio on is denoted as a *duty cycle*. Some existing algorithms transmit and listen with certain probabilities that are related to duty cycle [16,20,22], but they assume the network is fully connected, which is unrealistic in real IoT application. Some works focus on minimizing *discovery latency* [16,20], which is the time to discover all neighbors. However, they assume the nodes keep radios on all the time (duty cycle is 1) and these nodes would run out of energy quickly. Many deterministic protocols minimize discovery latency with a given duty cycle in [9,11,12,14,15,17,19]. Nevertheless, these methods only aim at two nodes, and they ignore the communication collisions among multiple nodes. By experiment, we find collisions caused by simultaneous transmissions result in a waste of time and energy. We considered a partially connected network with 1000 nodes and a probability of 0.5 for two nodes to be neighbors. We found that the collision slots of Hello [19], a deterministic protocol designed for two nodes, make up 99.8% of the time; while the collision slots of PND [18], a probabilistic protocol, make up 46.5% of the time. That means, because of collisions, existing works waste time and energy, and cannot achieve low latency and energy efficiency for the partially connected networks.

In this paper, we propose a low-latency and energy-efficient neighbor discovery algorithm called *Panacea* to address the above two challenges. Our algorithms take the sensors’ duty cycle into consideration and adopt the collision detection mechanism to shorten the discovery latency. We summarize our contributions as follows:
We propose the *Panacea* algorithm for a partially connected network which discovers all neighbors in O(nlnn) slots where *n* is the number of neighbors on average;With the collision detection hardware mechanism, we modify the algorithm, creating *Panacea*-WCD, which can alleviate the communication collisions;*Panacea* is suitable for both synchronous (i.e., all nodes are activated at the same time) and asynchronous (i.e., all nodes are activated at different time slots) scenarios.

We have conducted extensive simulations to evaluate the proposed algorithm. The results show that *Panacea* improves both latency and energy efficiency. For both synchronous and asynchronous scenarios, we analyze that the discovery latency can be bounded within O(nlnn) slots theoretically, and the simulation results also corroborate the analysis. Compared with Coupon [20] and PND [18], *Panacea* reduces the discovery latency by 60% and 92%, respectively, for synchronous scenarios, while reduces the discovery latency by 30% and 90% for asynchronous scenarios. *Panacea* could save 40% energy when it achieves similar discovery latency with Coupon. In addition, *Panacea* shows its best performance in regards to the trade-off between discovery latency and duty cycle, compared with Coupon, PND, Aloha-like [22], and Hello [19].

The rest of this paper is organized as follows. We review existing approaches in Section 2 and introduce the preliminaries in Section 3. Then, we propose the *Panacea* algorithm and analyze the efficiency in Section 4. With the collision detection mechanism, we propose the *Panacea*-WCD algorithm in Section 5 and present how the collision can be utilized in the algorithm. We implement *Panacea* and illustrate the comparison results in Section 6. Finally, we conclude this paper in Section 7.

## 2. Related Work

In recent years, wireless sensor networks have been extensively studied [1,3,4,5,6,7,8,23,24,25,26,27,28]; as a cornerstone of constructing wireless sensor networks, increasingly sophisticated protocols for neighbor discovery have been proposed. These neighbor-discovery protocols mainly fall into two categories. One category is the probabilistic methods, which is to exploit the probability theory to discover nodes within an expected time, statistically speaking. The other category is the deterministic methods, which is to utilize mathematical theorems to guarantee the discovery between two nodes.

In probabilistic algorithms, each node in the network transmits, listens, or sleeps with a certain probability at each time slot, and the sum of these three probabilities is 1. One of the traditional methods is Birthday protocols, seen in [16]. It utilizes random independent transmissions based on the birthday paradox (i.e., when there are 23 people, the probability that at least two have the same birthday exceeds 0.5), and saved a great deal of energy with a high expected proportion of discovered neighbors. This method was further studied as an Aloha-like algorithm in [20], based on the classical coupon collector’s problem. This algorithm first introduced the reception feedback with a collision detection mechanism, and mathematically showed that discovery latency was bounded in both scenarios, when nodes in the network are with or without a collision detection mechanism. Later on, a detailed physical layer mechanism was proposed in [29] for how nodes in receive mode detect the channel status, and methods at higher layers were also described based on the reception status information at transmitters. However, these methods only focused on improving the proportion of discovered neighboring nodes, ignoring the significance of saving energy.

Following that, similar and nicer probabilistic algorithms with a pre-defined duty cycle were proposed to save energy in [18,22]. However, these two methods only considered unrealistic fully connected networks, where every two nodes in the network are ideally connected. Besides, sophisticated hardware tools were used to reach a good tradeoff between latency and duty cycle in [30], but it required a complicated internal mechanism and increased the network cost. Recently, more methods targeted at collision problems are proposed in [31,32,33,34], but they introduced overhead to assist for packet collision indication, which adds to the complexity of networks.

In deterministic algorithms, radios are pre-scheduled to be “on” or “off” in each time slot based on some mathematical theorems to guarantee the discovery between every two neighbors. According to a survey in [35], three methods were usually adopted to guarantee the discovery, i.e., over-half occupation, quorum system, and co-prime. First, *over-half occupation* is to guarantee the overlap of active slots by keeping two nodes on for more than half of their slots. For example, given *n* slots in a period, if two nodes are active for at least $\frac{\left(n+1\right)}{2}$ slots, they must have some overlapping active slots. On the other hand, having more than half of the slots active means the energy usage is quite high. To reduce the excessive energy consumption, a more intelligent way is to divide a period into *r* cycles with *k* slots each cycle and allocate active slots in each cycle. This incentive was adopted in SearchLight [9]. Second, *quorum system* considers $m^{2}$ slots as an m×m slot matrix, where each node takes one row and one column slots as active. This idea, implemented in [11,14,15] ensures discovery due to the indispensable intersected active slots. Among these algorithms, only Hedis [11] supports asymmetric duty cycle. Third, *co-prime* takes advantage of the Chinese remainder theorem [36]. For example, methods like Disco [12], U-Connect [15], and Todis [11] guarantees the intersection of active slots by making nodes active at the product of preset numbers that are co-prime to one another. However, most of these deterministic algorithms are designed for two nodes, despite being applied to multi-node scenarios without devising any approaches to deal with collisions.

In a more realistic scenario, when nodes are in a large, partially-connected network, probabilistic algorithms have an edge over deterministic ones in terms of reducing the number of collisions. Given the duty cycle, we can adjust the probabilities of transmission and listening in each slot to alleviate collisions. Ideally, the goal of designing neighbor discovery protocols is to guarantee discovery within a reasonable amount of time while minimizing the number of active slots for each node to save energy. There are also some other algorithms that are designed for wireless sensor networks. Neighbor discovery algorithms are proposed with mobile sink nodes in [37,38] and a low latency protocol called Welcome is presented in [39] for mobile IoT devices. Some heuristic algorithms are also proposed with mobile sink nodes to improve the discovery performance in [40,41]. However, these algorithms are inapplicable since we do not consider mobile nodes in this paper.

## 3. Preliminaries

In this section, we introduce the system model and formulate the neighbor discovery problem for wireless sensor networks formally.

### 3.1. Sensor Node Model

Assuming that there are *N* sensor nodes in total that are deployed for a specific IoT application; we denote them as set $U=\{u_{1},u_{2},\ldots ,u_{N}\}$. Suppose each node $u_{i}$ has a unique identifier (ID) *i*. Time is assumed to be divided into slots of equal length $t_{0}$, which is sufficient to complete communications. All nodes can communicate through a specific channel for information exchanging. Suppose there are three states for each node: {Transmit,Listen,Sleep}, where Transmit means a node $u_{i}$ broadcasts (sends packets) on the channel; Listen means node $u_{i}$ listens on the channel and it can receive packets (message) from neighbors; and Sleep means node $u_{i}$ turns its radio off and does nothing to save energy. In each slot, a node can choose to be in any state by turning on or off its radio. For simplicity, we assume state transitions do not consume any time or energy, and only nodes that are transmitting or listening consume their battery power.

Denote the activation time of node $u_{i}$ as $t_{i}^{s}$, which implies the node does not work and stays in sleep mode until $t_{i}^{s}$. It is called a *synchronous scenario* if all deployed nodes are activated at the same time, i.e., $t_{i}^{s}=t_{j}^{s},\forall u_{i},u_{j}\in U$; otherwise it is called an *asynchronous scenario*. For a sufficiently long time from $t_{i}^{s}$ to *T*, denote the number of slots that node $u_{i}$ is in Transmit, Listen, or Sleep state as $T_{i}\left(T\right),T_{i}\left(L\right)$ or $T_{i}\left(S\right)$, respectively; $T_{i}\left(T\right)+T_{i}\left(L\right)+T_{i}\left(S\right)=T-t_{i}^{s}$. We define the duty cycle of the node as
(1)$$\theta_{i}=\frac{T_{i}\left(T\right)+T_{i}\left(L\right)}{T-t_{i}^{s}}.$$

For a specific IoT application, the duty cycle is normally defined in advance. In some hostile environment, it is costly to recharge the battery and thus the duty cycle would be very small, while the value could be large for some amicable applications.

### 3.2. Network Model

Most probabilistic works [16,20] only consider a fully connected network, where any two nodes are neighbors and they compose a fully connected network; this assumption is unrealistic. In this paper, we study the neighbor discovery process in a partially connected network, where some nodes are not neighbors, i.e., they are not connected directly for communication. We introduce two different network representations.

Considering a hostile environment where two nodes are neighbors with some probability, we use *neighboring matrix*
$M_{N\times N}$ to represent the neighboring relations. If node $u_{i}$ is a neighbor of $u_{j}$, we set $M_{ij}$ and $M_{ji}$ to be 1 (we assume the neighboring relation is undirected and $u_{j}$ is also a neighbor of $u_{i}$), else the value of the entrance is 0. We assume $u_{i}$ is a neighbor of $u_{j}$ with probability $p_{n}$ due to the obstacle or other blocking scenarios where $p_{n}\in \left(0,1\right)$. Clearly, the network is a fully connected one when $p_{n}=1$. In this paper, we consider a partially connected network where $p_{n}<1$. Therefore, a node would have $n=p_{n}\left(N-1\right)$ neighbors on average. For a large *N*, we can approximate $n\approx p_{n}N$ when it does not affect the analyses.

Considering another scenario where the nodes are deployed in a large field and the distance between two nodes may exceed the maximum radio range, we denote the distance of node $u_{i},u_{j}$ as $d(u_{i},u_{j})$ and two nodes are called neighbors ($M_{ij}=M_{ji}=1$) if $d(u_{i},u_{j})\leq \Delta$, where Δ is the maximum radio range two nodes can communicate. Then the network topology would be determined after the deployment of the sensor nodes. In this paper, we suppose the nodes are deployed by the uniform distribution in a large area *D*. If the area *D* is covered by a Δ×Δ rectangle, all nodes are located within their radio range and they compose a fully connected network. We assume a large area *D* that cannot be covered by such a rectangle and the nodes compose a partially connected network.

### 3.3. Collision Model

Generally speaking, when one node $u_{i}$ transmits through the channel, another node $u_{j}$ who listens on the channel simultaneously could receive the transmitted packet/message and decode the message successfully. However, if two or more nodes transmit concurrently on the channel, *communication collision* occurs and $u_{j}$ cannot decode the message correctly. Therefore, we assume a node $u_{j}$ can discover its neighbor $u_{i}$ only when $u_{j}$ listens on the channel while $u_{i}$ is $u_{j}$’s only neighbor who transmits.

In some design, communication collisions can be detected by hardware mechanism [18,20]. As described above, there are two situations for an unsuccessful discovery. The first scenario is no neighbor transmits while the second scenario is more than one neighbor transmit simultaneously. With the collision detection (CD) mechanism, a listening node can distinguish whether collisions occur or no neighbors is transmitting, apart from successful discovery. This CD mechanism enables the listening node to notify its transmitting neighbors of the transmission outcomes, and hence we can use this mechanism to design efficient algorithm.

### 3.4. Problem Definition

Neighbor discovery is not bidirectional, which means $u_{i}$ discovering $u_{j}$ is not equal to $u_{j}$ discovering $u_{i}$. We first define discovery latency between two nodes as follows.

**Definition** **1.**
*Discovery latency L(i,j) is defined as the duration from when the sensor node $u_{i}$ starts to when $u_{i}$ discovers its neighboring node $u_{j}$.*


Formally, suppose node $u_{i}$ listens in time slot *T* while its neighbor node $u_{j}$ transmits in slot $T_{j}$ (the only neighbor who transmits), we say $u_{i}$ discovers $u_{j}$ and the discovery latency can be computed as
(2)$$L\left(i,j\right)=T_{j}-t_{i}^{s}.$$

We define the discover problem for a specific node $u_{i}$ as follows.

**Problem** **1.**
*Considering an arbitrary node $u_{i}$ and the set of neighbors $NS\left(u_{i}\right)=\left\{u_{j}|M_{ji}=1\right\}$, design the algorithm for each node in each slot such that $u_{i}$ can discover all nodes in $NS(u_{i})$.*


We define the discovery latency of node $u_{i}$ as L(i), which represents the time to discover all neighbors:

(3)$$L\left(i\right)=\max_{u_{j}\in NS\left(u_{i}\right)}L\left(i,j\right)=\max_{u_{j}\in NS\left(u_{i}\right)}\left(T_{j}-t_{i}^{s}\right).$$

Since the sensor nodes may be activated in different time slots, we define the discovery latency of the network $L_{M}$ as the maximum discovery latency of all nodes:(4)$$L_{M}=\max_{u_{i}\in U}L\left(i\right).$$

The objective is to design an efficient algorithm that can discover the neighbors in a short time ($L_{M}$) under the pre-defined duty cycle in a partially connected network. We list the notations in Table 1.

## 4. *Panacea*: An Efficient Neighbor Discovery Algorithm

In this section, we describe *Panacea*, a low-latency, energy-efficient neighbor discovery algorithm for a partially connected network. To begin with, we assume the sensor nodes do not have a collision detection mechanism; we present the proposed *Panacea* algorithm and analyze the performance for both synchronous and asynchronous scenarios.

### 4.1. Algorithm Description

For any node $u_{i}$, suppose it has *n* neighbors on average and the pre-defined duty cycle of the node is $\theta_{i}$. When the nodes cannot detect communication collision, we introduce the idea of designing the probabilistic algorithm called *Panacea*-NCD (*Panacea* no collision detection).

Without a collision detection mechanism, node $u_{i}$ transmits with probability $p_{i}^{t}$, listens with probability $p_{i}^{l}$, and sleeps with probability $p_{i}^{s}$ in each time slot *t*. For each state, node $u_{i}$ takes the corresponding operations below.

If $u_{i}$ chooses state *Transmit*, it transmits a message containing its source ID on the channel;If $u_{i}$ chooses state *Listen*, it listens on the channel and decodes the source ID of the received message if it receives a message successfully;If $u_{i}$ chooses state *Sleep*, it does nothing.

We describe the algorithm formally as Algorithm 1. Each node $u_{i}$ repeats the algorithm every time slot until it discovers all neighbors. In each time, node $u_{i}$ generates a random value r∈(0,1) and takes corresponding actions when $r\leq p_{i}^{t}$, $p_{i}^{t}<r\leq \theta_{i}$, and $r>\theta_{i}$. The important part of the algorithm is to design the proper transmission probability $p_{i}^{t}$. It is obvious that $p_{i}^{t}+p_{i}^{l}+p_{i}^{s}=1$. Due to different applications, each node’s duty cycle is determined in advance and we denote it as $\theta_{i}$ for node $u_{i}$; since the duty cycle denotes the fraction of time slots that $u_{i}$ is transmitting or listening, hence $\theta_{i}=p_{i}^{t}+p_{i}^{l}$. Considering that nodes are comparatively well-distributed in the network, we suppose the probabilities of each node in each state are the same for simplicity. That is, ∀i∈[1,N], $p_{i}^{t}=p_{t}$, $p_{i}^{l}=p_{l}$, $p_{i}^{s}=p_{s}$, and $\theta_{i}=\theta$. To minimize the probability of collisions, we derive an approximation of the optimal transmission probability that can alleviate communication collisions to help achieve low discovery latency:(5)$$p_{i}^{t}=\frac{1}{n}.$$

We will show the procedure of deriving the transmitting probability in the following parts. As described in the algorithm, node $u_{i}$ selects the *Transmit* state with probability $p_{i}^{t}=\frac{1}{n}$ (Line 4–5), selects the *Listen* state with probability $\theta_{i}-p_{i}^{t}$ (Line 6–7), and selects the *Sleep* state with probability $1-\theta_{i}$ (Line 8–9). We analyze how to determine the transmission probability and show the efficiency of the algorithm.
**Algorithm 1***Panacea*-NCD ($u_{i},\theta_{i},n$). 1:$p_{i}^{t}\leftarrow \frac{1}{n}$2:**while** not terminate **do**3: r←random(0,1)4: **if**
$r\leq p_{i}^{t}$
**then**5:  node $u_{i}$ transmits on the channel6: **else if**
$p_{i}^{t}<r\leq \theta_{i}$
**then**7:  node $u_{i}$ listens8: **else**9:  node $u_{i}$ sleeps10: **end if**11:**end while**


### 4.2. Algorithm Analysis of Synchronous Scenario

We first analyze the algorithm’s efficiency of synchronous scenario where all nodes are activated simultaneously. We assume the network is represented by matrix $M_{N\times N}$ and any two nodes $u_{i},u_{j}$ are neighbors with probability $p_{n}\in \left(0,1\right)$. We first derive the algorithm’s discovery latency and derive the transmitting probability that minimizes the latency; then we show the upper bound of the discovery latency that can be utilized as a termination condition.

#### 4.2.1. Expectation Analysis of Discovery Latency

According to the *Panacea*-NCD algorithm, we can easily derive that the probability that node $u_{i}$ discovers a specific neighboring node $u_{j}$ successfully in a given slot is:(6)$$p_{suc}=p_{t}{\left(1-p_{t}\right)}^{n-1}\left(\theta -p_{t}\right).$$

This is because all nodes are activated at the same time and these nodes would select states independently. Node $u_{i}$ can discover $u_{j}$ only when $u_{j}$ selects the *Transmit* state with probability $p_{t}$, node $u_{i}$ selects the *Listen* state with probability $p_{l}=\theta -p_{t}$, while other n−1 neighbors select the *Sleep* or *Listen* state with probability $p_{l}+p_{s}=1-p_{t}$.

By maximizing $p_{suc}$, we can alleviate the maximum amount of collisions. Thus, we derive the transmission probability $p_{t}$ that maximizes $p_{suc}$ with the derivative
(7)$$p_{t}=\frac{\theta n+2-\sqrt{4+{\left(\theta n\right)}^{2}-4\theta }}{2(1+n)}\approx \frac{1}{n}.$$

Substituting this into the formulation, the probability that node $u_{i}$ discovers a specific neighbor successfully in a given slot is $p_{suc}\approx \frac{\theta }{en}$. We approximate these equations when *n* is a large value and the approximation does not impact the discovery latency largely. When *n* is small, we can compute the exact value of $p_{t}$ and derive the discovery probability.

We next show that maximizing $p_{suc}$ helps to achieve the lowest expected latency. Let *W* be a random variable that denotes the time a node spends discovering all neighbors. Considering any node, we define the time spent in discovering a new neighbor (the *j*-th node) after it discovered j−1 neighbors to be $W_{j}$, which follows a geometric distribution with parameter $p_{suc}\left(j\right)$: $p_{suc}\left(j\right)=\left(n-j+1\right)p_{suc}$.

Hence, the expectation of $W_{j}$ can be computed as:(8)$$E\left[W_{j}\right]=\frac{1}{p_{suc}\left(j\right)}=\frac{1}{\left(n-j+1\right)p_{suc}}.$$

Clearly, $W=W_{1}+W_{2}+...+W_{n}$ which implies the time node $u_{i}$ discovers all neighbors; the expectation of *W* can be formulated as
(9)$$E\left[W\right]=\sum_{j=1}^{n}E\left[W_{j}\right]=\frac{1}{p_{suc}}H_{n}$$
where $H_{n}$ is the *n*-th harmonic number, i.e., $H_{n}=\ln n+\Theta \left(1\right)$. By maximizing $p_{suc}$, the lowest expected discovery latency becomes:(10)$$E\left[W\right]\approx \frac{ne}{\theta }\left(\ln n+\Theta \left(1\right)\right)=\Theta \left(\frac{n}{\theta }\cdot \ln n\right).$$

When the duty cycle is a pre-defined parameter, the expected discovery latency can be bounded as Θ(n·lnn). Therefore, we can conclude the following theorem:

**Theorem** **1.***The* Panacea*-NCD algorithm ensures a node $u_{i}$ can discover all its n neighboring nodes within Θ(n·lnn) time slots with a high probability for a synchronous scenario.*

#### 4.2.2. Upper Bound Analysis of Discovery Latency

We show that the discovery latency is not likely to be much larger than its expectation, which can be utilized as the termination condition in Algorithm 1.

Recall the definition of $W_{i}$ in the above analysis; if $W_{i}$ is given, the value of $W_{j}$ will not be affected for i<j. That is, i≠j, $W_{i}$, and $W_{j}$ are independent and they satisfy $P\left(W_{j}=w_{j}|W_{i}=w_{i}\right)=P\left(W_{j}=w_{j}\right)$. Since $W_{j}$ follows geometric distribution and $Var\left[W_{j}\right]=\frac{1-p_{j}}{p_{j}^{2}}$, the variance of *W* can be computed as:(11)$$\begin{array}{c} Var\left[W\right]=\sum_{j=1}^{n}Var\left[W_{j}\right]\leq \frac{\pi^{2}}{6p_{suc}^{2}}-\frac{H_{n}}{p_{suc}}. \end{array}$$

According to the *Chebyshev’s inequality*, we can derive the probability that the discovery latency exceeds the expected time by two times, as
(12)$$\begin{array}{c} P\left[W\geq 2E\left[W\right]\right]\leq \frac{Var[W]}{E{\left[W\right]}^{2}}\leq \frac{\pi^{2}}{6H_{n}^{2}}-\frac{p_{suc}}{H_{n}}. \end{array}$$

For a large *n* where n→∞, and P[W≥2E[W]] is close to 0. That is, the time for a node to find all neighbors is very likely to be smaller than 2 times the expected latency. Therefore we derive W=O(n·lnn). We can conclude the corollary as follows.

**Corollary** **1.***The* Panacea*-NCD ensures a node $u_{i}$ can discover all its n neighboring nodes in O(nlnn) time slots a with high probability for a synchronous scenario*
*when n→∞*.

### 4.3. Algorithm Analysis of Asynchronous Scenario

The proposed *Panacea*-NCD algorithm is suitable for asynchronous scenario where the sensor nodes are activated in different time slots. For node $u_{i}$, denote the maximum activation time offset between $u_{i}$ and any neighbor is $\delta =\max_{u_{j}\in NS\left(u_{i}\right)}\left(t_{i}^{s}-t_{j}^{s}\right)$. Denote the latency that $u_{i}$ finds a new neighbor after it discovered j−1 neighbors as $W_{j}^{\prime }$, and the latency that $u_{i}$ finds all neighbors as $W^{\prime }$.

It is obvious that the start time offset δ will not affect the latency, which implies
(13)$$E\left[W_{j}^{\prime }\right]=E\left[W_{j}\right]+\delta .$$

Hence, we derive the expected latency for a node discovering all its neighbors as
(14)$$E\left[W^{\prime }\right]\approx \frac{e}{\theta }n\cdot \ln n+\left[\Theta \left(1\right)+\delta \right]n=\Theta \left(n\cdot \ln n\right).$$

We can conclude that the expected discovery latency for an asynchronous scenario is δn slots larger than that for a synchronous scenario. When δ is a constant value, the expected discovery latency is also Θ(n·lnn). Similarly, we can also derive the probability that the discovery latency is 2 times larger than the expectation, as $P[W^{\prime }\geq 2E\left[W^{\prime }\right]]$ is still close to 0; hence the discovery latency can be bounded by O(n·lnn) with high probability. Combining these, we conclude the following theorem:

**Theorem** **2.***The* Panacea*-NCD algorithm ensures a node $u_{i}$ can discover all its n neighboring nodes within Θ(n·lnn) slots in expectation and in O(n·lnn) slots with a high probability for an asynchronous scenario*
*when n→∞*.

**Remark** **1.**
*We assume $p_{n}\in \left(0,1\right)$ in our analyses for the following reasons. When $p_{n}=0$, it implies all nodes are completely separated and no node is connected with even one neighboring node; then it is meaningless to analyze the algorithm performance. When $p_{n}=1$, the network is fully connected which implies any node can communicate with all other nodes directly. The analyses can also be adopted to the situation. Since we define the network as a partially connected one, we assume $p_{n}<1$.*


### 4.4. Analysis for Uniform Distribution

Uniform distribution is used in most deployments of wireless networks. For instance, to monitor an unknown area, many sensors are deployed uniformly to collect information, such as temperature and humidity [42]. The nodes are evenly deployed and the density function can be formulated as:(15)$$f\left(x,y\right)=\left\{\begin{array}{cc} \frac{1}{A} & (x,y)\in D\\ 0 & (x,y)\notin D \end{array}\right.$$
where *A* is the area of *D*. For any node $u_{i}$, denote the range of $u_{i}$’s neighbors’ positions as $R_{i}$ and any neighboring node with coordinate $\left(x,y\right)\in R_{i}$ suits ${\left(x-x_{i}\right)}^{2}+{\left(y-y_{i}\right)}^{2}\leq \Delta^{2}$, where $\left(x_{i},y_{i}\right)$ is $u_{i}$’s position. Then, node $u_{i}$’s expected number of neighbors can be computed as
(16)$$n=N{\;\int \int \;}_{R_{i}}f\left(x,y\right)\;dx\;dy-1≃\frac{N\pi \Delta^{2}}{A}.$$

Combining this in the *Panacea*-NCD’s design, we can still derive the upper bound of discovery latency to be O(n·lnn) when *n* is large and compute the latency precisely.

We analyze the algorithm when the nodes have nearly the same number of neighbors for different network representations (including the uniform distribution). When the nodes obey some other distribution, the nodes may have a different number of neighbors and it would be an interesting future work. In addition, the analyses of the expectation and the upper bounds are derived when *n* is large. When *n* is small, these bounds may not hold but they show the increasing trend when *n* increases.

## 5. Panacea-WCD: Neighbor Discovery with Collision Detection

In this section, we propose *Panacea*-WCD algorithm, a novel random algorithm that achieves low latency for a given duty cycle when the nodes can detect collision.

### 5.1. Algorithm Description

If nodes could identify collisions, the transmitting node(s) could be notified whether transmissions are successful by their listening neighbors. We describe *Panacea*-WCD in Algorithm 2 and it works as follows:

Each time slot is divided into two sub-slots. In the first sub-slot, nodes transmit, listen, or sleep and act in response to one of the three states. In the second sub-slot, nodes notify their neighbor(s) and maintain *DiscoveredList* to record the discovered nodes.

Initially, each node $u_{i}$ sets $k_{i}=0$, and α is a constant. For each time slot *t*,

(1)In the first sub-slot, node $u_{i}$ transmits a message containing its source node ID with probability $p_{i}^{t}=\frac{1}{n+\alpha k_{i}}$ (Line 4), listens with probability $p_{i}^{l}=\theta_{i}-p_{i}^{t}$, and sleeps with probability $p_{i}^{s}=1-\theta_{i}$;(2)In the second sub-slot, there are three sub-cases:
-If $u_{i}$ selects the Listen state in the first sub-slot: if $u_{i}$ receives a message successfully, $u_{i}$ decodes and records the source ID in the message. If the ID does not belong to $u_{i}$’s *DiscoveredList*, $u_{i}$ adds the ID to *DiscoveredList*, and deterministically transmits a message on the channel (a bit is sufficient) in the second sub-slot;-If $u_{i}$ selects the Transmit state in the first sub-slot: if $u_{i}$ detects energy (a message or a collision by multiple messages), $u_{i}$ is notified the successful transmission and sets $k_{i}:=k_{i}+1$, otherwise $u_{i}$ regards its transmission as unsuccessful in the first sub-slot;-If $u_{i}$ selects the Sleep state in the first sub-slot: $u_{i}$ does nothing in the second sub-slot.

The core of *Panacea*-WCD is that once $u_{i}$ discovers a new node $u_{j}$ that does not belong to its *DiscoveredList*, $u_{i}$ adds $u_{j}$’s ID to its *DiscoveredList* and notifies $u_{j}$ of the successful discovery. In this successful scenario, $u_{i}$’s feedback in the second sub-slot can be 1-bit. Thus, the second sub-slot is much smaller than the first one, and it only introduces a small overhead.
**Algorithm 2** Panacea-WCD($u_{i},\theta_{i},\alpha ,n$). 1:$k_{i}\leftarrow 0$, DiscoveredList←{}2:**while** not terminate **do**3: $p_{i}^{t}\leftarrow \frac{1}{n+\alpha k_{i}}$, r←random(0,1)4: **if**
$r<p_{i}^{t}$
**then**5:  node $u_{i}$ transmits in the first sub-slot6:  **if** detects energy in the second sub-slot **then**7:   SuccessTransmission←True, $k_{i}\leftarrow k_{i}+1$8:  **else**9:   SuccessTransmission←False10:  **end if**11: **else if**
$p_{i}^{t}<r<\theta_{i}$
**then**12:  node $u_{i}$ listens in the first sub-slot13:  **if** Successful reception in the first sub-slot **then**14:   node $u_{i}$ decodes the message and the source ID (srcID)15:   **if** srcID not in DiscoveredList
**then**16:    DiscoveredList=DiscoveredList⋃{srcID}17:   **end if**18:  **end if**19: **else**20:  node $u_{i}$ sleeps in the first sub-slot21: **end if**22:**end while**


### 5.2. Algorithm Analysis of Synchronous Scenario

We also analyze the discovery latency from both expectation and upper bound aspects.

#### 5.2.1. Expectation Analysis of Discovery Latency

Similar to *Panacea*-NCD, we suppose the latency to find a new neighbor after discovered j−1 neighbors is $W_{j}$. To simplify this analysis, we consider a node $u_{i}$ discovering neighbors at an average level, where the number of discovered neighbors follows a normal distribution. That is, some nodes that discover neighbors faster and slower tend to compensate for the number of discovered neighbors of each other. In other words, we have a global transmission probability $p_{t}\left(j\right)$ in $W_{j}$, which is
(17)$$p_{t}\left(j\right)=\frac{1}{n+\alpha (j-1)}.$$

In a given time slot, the probability that $u_{i}$ discovers a new neighbor after it discovered j−1 neighbors is
(18)$$p_{suc}\left(j\right)=p_{t}\left(j\right){\left(1-p_{t}\left(j\right)\right)}^{n-1}\left(\theta -p_{t}\left(j\right)\right)\left(n-j+1\right).$$

As $W_{j}$ follows geometric distribution with parameter $p_{suc}\left(j\right)$, we choose α close to 1, $\frac{1}{n+\alpha (j-1)}\ll \theta$, and the latency expectation that $u_{i}$ discovers all neighbors $W=\sum_{i=1}^{n}W_{i}$ can be computed as:(19)$$\begin{array}{cc} E[W] & =\sum_{j=1}^{n}E\left[W_{j}\right]=\sum_{j=1}^{n}\frac{1}{p_{suc}\left(j\right)}. \end{array}$$

Because $\sum_{j=1}^{n}\frac{n+\alpha (j-1)}{n-j+1}=\left(\alpha +1\right)nH_{n}-\alpha n$ where $H_{n}$ is *n*-th *harmonic number*, we can obtain:(20)$$\frac{n}{\theta }\left[\left(\alpha +1\right)\ln n+\Theta \left(1\right)\right]\leq E\left[W\right]\leq \frac{en}{\theta }\left[\left(\alpha +1\right)\ln n+\Theta \left(1\right)\right].$$

When θ is determined in advance, we can conclude the theorem as

**Theorem** **3.***The* Panacea*-WCD algorithm ensures a node $u_{i}$ can discover its all n neighboring nodes within Θ(n·lnn) time slots in expectation for synchronous scenario.*

#### 5.2.2. Upper Bound Analysis of Discovery Latency

We show that the latency is not likely to be much larger than its expectation. For i≠j, $W_{i}$ and $W_{j}$ are independent by definition. As $Var\left[W_{j}\right]=\frac{1-p_{suc}\left(j\right)}{p_{suc}^{2}\left(j\right)}$ for geometric distribution with parameter $p_{suc}\left(j\right)$, we obtain:(21)$$Var\left[W\right]=\sum_{j=1}^{n}Var\left[W_{j}\right]=\sum_{j=1}^{n}\frac{1}{p_{suc}^{2}\left(j\right)}-\sum_{j=1}^{n}\frac{1}{p_{suc}\left(j\right)}.$$

We know that $\sum_{j=1}^{n}\frac{1}{p_{suc}^{2}\left(j\right)}\leq \frac{e^{2}}{\theta^{2}}\left[\alpha^{2}n-2\alpha n\left(1+\alpha \right)H_{n}+\frac{\pi^{2}}{6}\left(1+\alpha^{2}\right)n^{2}\right]$; according to *Chebyshev’s inequality*, we derive:(22)$$\begin{array}{c} P\left[W\geq 2E\left[W\right]\right]\leq \frac{e^{2}\pi^{2}\left(1+\alpha^{2}\right)/6}{{\left[\left(1+\alpha \right)H_{n}-\alpha \right]}^{2}}-\frac{\theta /(en)}{\left[\left(\alpha +1\right)H_{n}-\alpha \right]}. \end{array}$$

The probability is close to 0 when *n* is large (n→∞). Hence, the latency is not likely to be 2 times larger than the expectation. Therefore, we conclude the corollary as:

**Corollary** **2.***The* Panacea*-WCD ensures a node $u_{i}$ can discover its all n neighboring nodes in O(nlnn) time slots with high probability for synchronous scenario*
*when n→∞*.

### 5.3. Algorithm Analysis of Asynchronous Scenario

For *Panacea*-WCD, we can also derive Equation (Equation 13) of latency expectation to discover a new neighbor after the discovered j−1 neighbors. The expected latency for a node discovering all its neighbors is
(23)$$\frac{\left(\alpha +1\right)nH_{n}}{\theta }+\left(\delta -\alpha /\theta \right)n\leq E\left[W\right]\leq \frac{e}{\theta }\left[\left(\alpha +1\right)nH_{n}\right]+\left(\delta -\alpha e/\theta \right)n.$$

We can derive $E\left[W^{\prime }\right]=\Theta \left(n\cdot \ln n\right)$ for asynchronous scenario. Similarly to *Panacea*-NCD, the probability that the discovery latency is 2 times larger than the expectation is still close to 0. Hence, we conclude the following theorem:

**Theorem** **4.***The* Panacea*-WCD algorithm ensures a node $u_{i}$ can discover its all n neighboring nodes within Θ(n·lnn) slots in expectation and in O(n·lnn) slots with a high probability for an asynchronous scenario*
*when n→∞*.

Notice that, when the proposed algorithm is deployed in realistic network technologies such as Zigbee and BLE Mesh, we have to adjust the algorithm if their collision detection mechanism is applicable. For example, Zigbee utilizes CSMA/CA to reduce collision, which is difficult to couple with *Panacea*-WCD, while BLE Mesh finishes events with a random delay, we may adjust the transmit and listen operations.

**Remark** **2.***The* Panacea *algorithm*
*(including both Algorithm 1 and Algorithm 2) needs to know the average number of neighbors in advance. The introduced two network representations help identify this value. In practice, the sensors are deployed artificially or by some distribution (such as the uniform distribution in Section 4.4); it is reasonable to derive the number of neighbors beforehand. It would be an interesting future work to relax the limitation.*


## 6. Evaluation

We implemented *Panacea* in C++ and evaluated the algorithms in a cluster with four nodes, each with an Intel(R) Xeon(R) CPU E5-2640 v4 @ 2.40 GHz, 12 cores, 128 GB memory, and 2 GB spinning disk. We used two different initialization methods to generate the networks. There are *N* sensor nodes in total and we set the maximum activation offset δ=1000. The first method generates the neighboring matrix $M_{N\times N}$ with probability $p_{n}$, which implies node $u_{i}$ is the neighbor of any other node $u_{j}$ (i.e., $M_{ij}=M_{ji}=1$) with probability $p_{n}$. The second method assumes all nodes are deployed in an area of size 100×100 m by the uniform distribution and the communication range is Δ=10 m. Any two nodes $u_{i},u_{j}$ satisfying $d(u_{i},u_{j})\leq \Delta$ are neighbors in the network. These two methods can generate the networks more complicated and realistic than that in [9,11,12,14,15,16,17,18,19,20,22].

We evaluate discovery latency of *Panacea*, Coupon [20], Aloha-like [22], and PND [18] in the generated partially-connected networks in both synchronous and asynchronous scenarios. Since Coupon [20] and PND [18] do not consider duty cycle, we evaluate the tradeoff of duty cycle and discovery latency among *Panacea*, Aloha-like [22], and Hello [19]. We also compare the evaluation results with the theoretical analyses. The expected discovery latency of *Panacea*-NCD is derived as Equation (Equation 10) while the expected latency of *Panacea*-WCD is bounded by Equation (Equation 20). We set the length of each time slot as $t_{0}=20$ ms and the nodes are activated randomly within time [0,δ] for asynchronous scenario; the results are based on 1000 separate runs.

### 6.1. Partially- and Fully-Connected Networks

To begin with, we evaluate the performance of the algorithms in different settings. Though *Panacea* is proposed for partially connected networks, it can be also adopted in a fully-connected network. As shown in Figure 1 and Figure 2, when the network is partially connected ($p_{n}=0.1,0.5$), the algorithms use less time to discover the neighbors (y-axis, number of slots) compared to the fully connected setting when there are N=1000 node in total. Among them, *Panacea* shows superior performance.

### 6.2. Comparison of Discovery Latency

We evaluate the performance of the proposed algorithms in different settings. In the first place, we generate the network with N=1000 nodes and $p_{n}=0.1$. We compare *Panacea* with Coupon and PND when the nodes always turn their radios on (i.e., the duty cycle is 1). As shown in Figure 3a where *y*-axis represents discovery latency, i.e., the number of slots, *Panacea*-NCD could achieve the lowest discovery latency when the nodes do not have a collision detection mechanism, reducing time by about 60% and 92% when compared to Coupon and PND, respectively. *Panacea*-WCD also has the best performance when the node can detect collision as illustrated in Figure 3b; it reduces latency by about 30% and 90% compared to Coupon and PND, respectively. Coupon can achieve good performance in a fully-connected network, but it works worse than *Panacea* in a partially-connected network. We do not compare the Aloha-like algorithm because it is the same with Coupon when the duty cycle is 1. By setting different duty cycles in our algorithm, *Panacea* could save energy by 40% when it achieves similar discovery latency with Coupon.

We also evaluate the algorithms when the number of nodes increases from 100 to 1000. As depicted in Figure 4, when the network size increases (x-axis), the discovery latency (y-axis, number of slots) increases for all algorithms. This is because each node has more neighbors to discover and the communication collision occurs more often. We compare *Panacea* to Coupon and PND for four different scenarios regarding whether the nodes are activated synchronously and whether they have a collision detection mechanism. The results show that our algorithms have the best performance, achieving lowest discovery latency when the number of nodes increases. Figure 4a,b also depicts the derived expected discovery latency of *Panacea*-NCD, and the results show that the evaluations corroborate the theoretical analyses. Regarding *Panacea*-WCD, Figure 4c,d shows the upper bound and lower bound of the expected discovery latency and the evaluation results also corroborate the analyses.

When the network is generated according to the nodes’ positions, we compare the neighbor discovery algorithms in Figure 5. When the nodes are deployed according to the uniform distribution and the number of nodes *N* increases from 100 to 1000, we also evaluate their performance for four different scenarios; the results show that the discovery latency increases when *N* increases and *Panacea* could achieve the lowest discovery latency among them. In addition, comparing with the expected discovery latency of *Panacea*, the evaluation results corroborate the theoretical analyses in Equation (Equation 10) and Equation (Equation 20).

As analyzed, *Panacea* could achieve the lowest discovery latency because Coupon and PND cannot handle communication collision efficiently, while *Panacea* could reduce discovery latency by handling the collision in an appropriate way.

### 6.3. Comparison of Discovery Rate

We evaluate the discovery rate of the algorithms which is defined as the percentage of discovered neighbors. We first evaluate *Panacea*-WCD for a dense network $p_{n}=0.5$, N=1000 and the duty cycle is θ=0.5. Figure 6 shows the discovery rate (y-axis) increases as time (x-axis) increases. In the synchronous scenario, *Panacea*-WCD could reach 100% discovery rate, while Coupon reaches 79.2% and PND reaches 84.6%, respectively. This is because Coupon and PND simply regard the case of no collision as a success, ignoring that some neighbors of transmitting nodes fail to discover in the *Sleep* state. The results show the similar phenomenon in the asynchronous scenario.

We evaluate the discovery rate when N=1000,θ=0.5 and all nodes are uniformly distributed. As shown in Figure 7, *Panacea* achieves higher discovery rate than others; both *Panacea*-NCD and *Panacea*-WCD could reach 100% discovery rate. Coupon could reach (nearly) 100% discovery rate in some scenarios when the network is sparser than that in Figure 6; this is because Coupon is more applicable to a sparse network.

These figures validate that *Panacea* could handle communication collision in the algorithm design and thus it achieves 100% discovery rate.

### 6.4. Tradeoff: Discovery Latency v.s. Duty Cycle

Since different applications could set different duty cycle values, we evaluate the tradeoff between discovery latency and duty cycle. When $N=1000,p_{n}=0.5$, Figure 8 demonstrates that *Panacea* has a better tradeoff between discovery latency (y-axis, number of slots) and duty cycle (x-axis); the discovery latency is proportional to $\frac{1}{\theta }$ for *Panacea*, which corroborates our analyses. The evaluated discovery latency of *Panacea*-NCD fits quite well with the expected discovery latency, while the evaluated latency of *Panacea*-WCD is also located within the lower bound and the upper bound of the expected latency. PND and Hello have higher latency, because PND’s collision slots make up 46.5% of the time, and Hello’s collision slots make up 99.8% of the time. The *power–latency product* [15] measurement is the product of the average power consumption with discovery latency. The power–latency product of *Panacea* is 9213, while that of Aloha-like is 20.3% larger, and PND is 2.69 times larger than *Panacea* in the synchronous scenario. The PND algorithm’s performance is unstable when the duty cycle increases, this is because PND suffers from irregular collisions and the algorithm varies the sending probability according to different scenarios. In addition, we only measured the average discovery latency of 1000 separate runs.

### 6.5. Network Density

Finally, we evaluate the performance of *Panacea* when the network becomes denser. We fix N=1000 and increase $p_{n}$ from 0.1 to 0.9; as shown in Figure 9, the discovery latency (y-axis, number of slots) increases for both scenarios (without or with collision detection). This is because collisions take place more frequently in denser networks. We depict three curves for different duty cycles (0.1,0.3,and 0.5, respectively); the results show that higher duty cycle contributes to lower latency, since discovery latency is proportional to $\frac{1}{\theta }$. The results validate that the discovery latency of *Panacea* is proportional to $\frac{1}{\theta }$ and that a larger $p_{n}$ leads to larger latency due to the collisions. The evaluated results also corroborate the theoretical analyses of the algorithm.

From these figures, the proposed *Panacea* algorithm can achieve a low discovery latency, a high discovery rate, and good tradeoff between discovery latency with duty cycle.

## 7. Conclusions

In this paper, we propose *Panacea* a low-latency and energy-efficient neighbor discovery algorithm for partially connected wireless sensor networks. First, we present *Panacea*-NCD for nodes without collision detection mechanism and we analyze that the discovery latency is bounded within O(n·lnn) time slots for any pre-defined duty cycle when *n* is large. Then, we modify this method in *Panacea*-WCD when nodes can detect collision. The discovery latency of *Panacea*-WCD is also bounded within O(n·lnn) slots for a large *n*. Our algorithms can be efficient in both synchronous and asynchronous scenarios because we alleviate collisions as much as possible. Furthermore, we verified our theoretical analyses by simulations and the results corroborate our analyses. Still, *Panacea* needs to know an average number of neighbors beforehand and that the nodes have the same number of neighbors, it would be important and interesting to relax such a priori knowledge in the future. In addition, we will evaluate the algorithms’ performance in further practical applications.

## Figures and Tables

**Figure 1 sensors-20-00657-f001:**
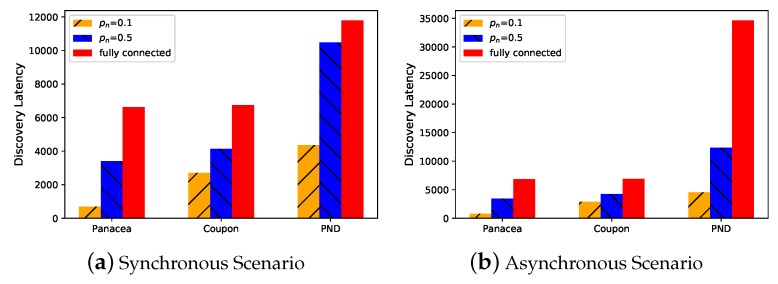
The neighbor discovery algorithms work better in partially connected networks and *Panacea*-NCD (*Panacea* no collision detection) outperforms the other algorithms.

**Figure 2 sensors-20-00657-f002:**
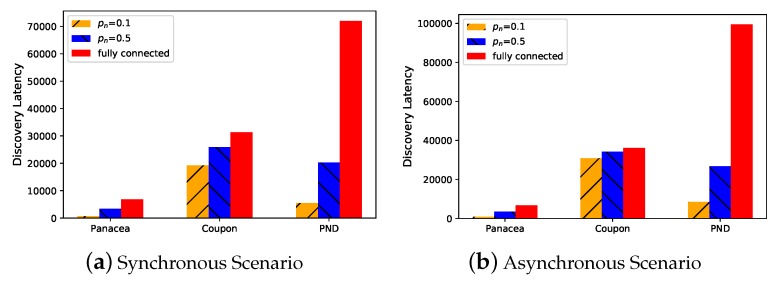
The neighbor discovery algorithms work better in partially connected networks and *Panacea*-WCD outperforms the other algorithms.

**Figure 3 sensors-20-00657-f003:**
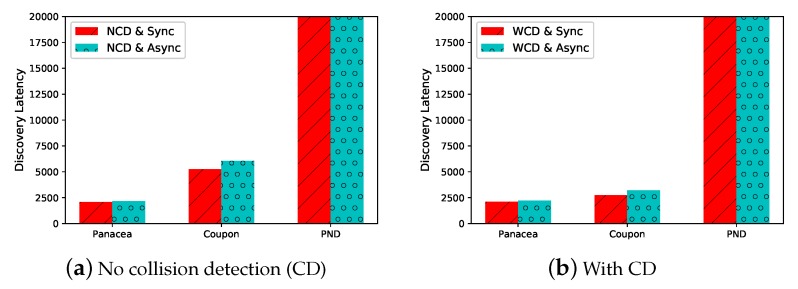
*Panacea*-NCD and *Panacea*-WCD have the lowest discovery latency under both synchronous and asynchronous scenarios when $p_{n}=0.1,N=1000$.

**Figure 4 sensors-20-00657-f004:**
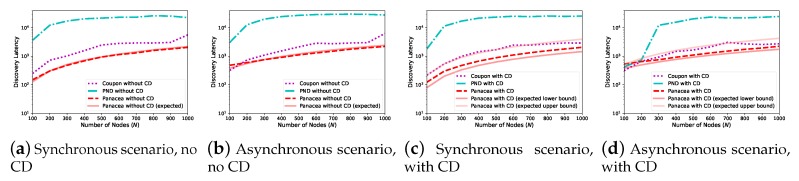
*Panacea*-NCD and *Panacea*-WCD achieve lowest discovery latency for both synchronous and asynchronous scenarios when $p_{n}=0.1$ and *N* increases from 100 to 1000.

**Figure 5 sensors-20-00657-f005:**
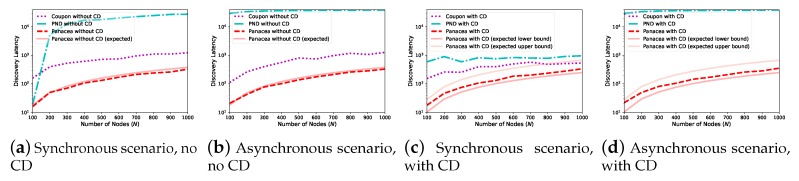
*Panacea*-NCD and *Panacea*-WCD achieve the lowest discovery latency for both synchronous and asynchronous scenarios under the uniform distribution.

**Figure 6 sensors-20-00657-f006:**
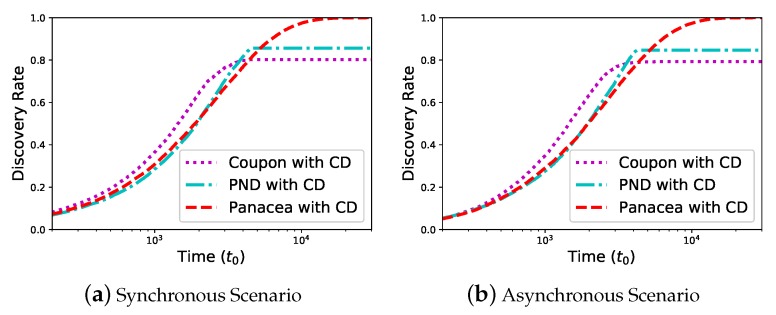
Panacea-WCD achieves higher discovery rate with CD.

**Figure 7 sensors-20-00657-f007:**
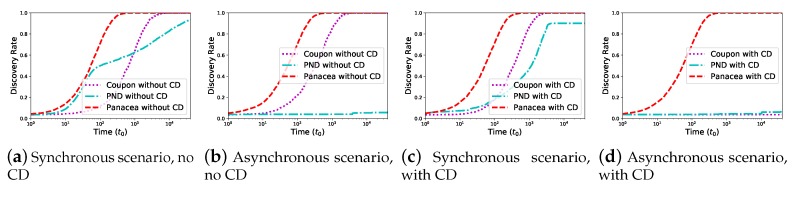
*Panacea*-NCD and *Panacea*-WCD achieve higher discovery rate for both synchronous and asynchronous scenarios under the uniform distribution.

**Figure 8 sensors-20-00657-f008:**
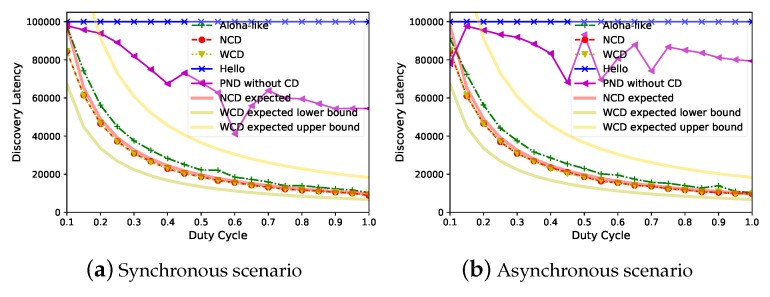
*Panacea* has better tradeoff between duty cycle and latency for both synchronous and asynchronous scenarios.

**Figure 9 sensors-20-00657-f009:**
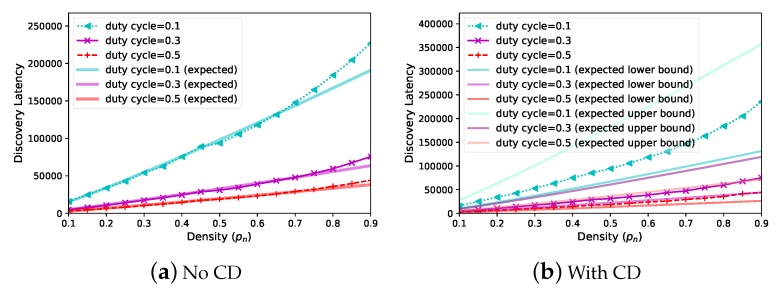
Discovery latency increases when the network becomes denser.

**Table 1 sensors-20-00657-t001:** Notations for neighbor discovery.

Notation	Description
*U*	The set of all sensor nodes $U=\{u_{1},u_{2},\ldots ,u_{N}\}$
$u_{i}$	Sensor node $u_{i}$ with ID *i*
*N*	The number of nodes in the network is *N*
$t_{0}$	The length of a time slot is $t_{0}$
$t_{i}^{s}$	The activation time of sensor node $u_{i}$
$\theta_{i}$	The duty cycle of sensor node $u_{i}$
*M*	Neighboring matrix, $M_{ij}=1$ means $u_{i}$ and $u_{j}$ are neighbors
$p_{n}$	A node is the neighbor of another with probability $p_{n}$
$d(u_{i},u_{j})$	The distance between sensor nodes $u_{i}$ and $u_{j}$
Δ	The maximum communication (radio) range
*D*	A large monitoring area
L(i,j)	The discovery latency that node $u_{i}$ discovers node $u_{j}$
$NS(u_{i})$	The set of node $u_{i}$’s neighboring nodes
L(i)	The discovery latency that node $u_{i}$ discovers all neighbors
$L_{M}$	The discovery latency of the network

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
