# Peer review of "A Low-Latency and Energy-Efficient Neighbor Discovery Algorithm for Wireless Sensor Networksâ€"

_sensors, 2020, doi:10.3390/s20030657_

Round 1
Reviewer 1 Report
This paper proposed a neighbor discovery algorithm. However, the motivation cannot convince me. Furthermore, some importance new results are not included in the paper. Other comments are listed as follow.
This paper has some symbol errors which is not friendly for reading. For example, ”we summarize our contributions are as follows”, ”We define the discovery latency of L(i) node ui”, ”we can also derive Eqn. (5)”, “When the nodes cannot detect communication detection”. The network is partially connected, which may have an influence on the average number of node. some theoretical analyses should be added. Cannot the probability be 0 or 1? In Alg.1, the symbol “r” should be explained. This paper proposes some algorithms for a partially connected network, while the algorithms are well presented, some details about partially network should be added. The equations in this paper should all be numbered.Author Response
Thanks for your comments! We modified the manuscript and respond your comments as the attached file. Thanks!

Reviewer 2 Report
The paper focuses on the neighbor discovery problem in a partially connected WSN networks. The paper is an enhanced version of the paper published on the WCNC conference. The main contribution are: i) the proposal for neighbor discovery algorithm, called Panacea, proposed for a network with and without collision detection algorithm, ii) the model of neighbor discovery process allowing to derive performance characteristics, iii) simulation experiments focused on Panacea performance evaluation.
The main drawbacks are the following:
1) The main technical contribution, i.e. Panacea algorithm, has been already published in the WCNC conference proceedings. The extensions in this papers cover: i) more detailed description of the algorithm and model, ii) enhanced state of the art analysis, and iii) new experiments focused on performance evaluation. It seems reasonable to enhance the paper scope of some new features of the proposed Panacea algorithm.
2) The presented model has not been validated at all. It is strongly required to compare results coming from the model with the simulation results. Otherwise, it is not justified why the description of the model is included in the paper. Moreover, the validation should assess the accuracy of the models. Please extend evaluation section by including comparison of the results coming from the proposed model and from simulations.
3) Some assumptions of the model are questionable or at least need deeper justification.
- in sec. 3.2, the average number of neighbors n=pn*N , seems inadequate because a given node is not the neighbor of itself. So the correct number of neighbors for pn=1, should be N-1 instead of N. This has also impact on other equations in the paper.
- in sec. 4.1, the algorithm 1 is repeated until it discovers all neighbors. Dose it means that the number of neighbors known a'priori?
- in sec. 4.1, prob. pt is fixed a'priori? How it is fixed? One sentence says that duty cycle is different for particular node, while next sentences say it is the same for all nodes? Please make it clear how it is related with nodes activity.. BTW, the assumption that all sensors have exactly the same characteristic is very strong. It needs more explanation.
4) There are some doubts about the model
- in sec. 4.2, there are some approximations in equations, but there is no discussion about consequences. Approximations are upper/lower limits? the conclusion about upper limit of discovery latency is true if n-> infinity. However is n is rather small, duty cycle has impact, hasn't it? (this is also one of the conclusion from Fig. 7 )
- I have doubts about usefulness of proposed upper bounds, that seems far away as typically Chebyshev's bound is far away. Please evaluate them and provide numbers how far it is from actual values. I'm not convinced about your conclusion "We show the discovery latency is bounded by O(nlnn) for any duty cycle (the percentage time to turn on the radio), where each node has n neighbors in average. For nodes that can detect collisions, we then present Panacea-WCD which also bounds the latency within O(nlnn) slots." Probably, this is true but the bounds is too far away from actual values, so it seems impractical. Please reconsider these conclusions taking into account results form Fig 7. ( I see here a bit contradiction).
5) Please discuss about deployment of proposed discovery algorithm in realistic network technologies as e.g. Zigbee, BLE MESH, etc. I mean that a given technology has some constraints that may influence proposed solution. This discussion is especially required for PanaceaWDC.
6) There should be included explanation about obtained results not just description. The most important is explanation why there are such differences between proposed Panacea and other analysed algorithms.
7) There are visible irregularities in Fig. 6. Please explain why?
8) Some minor editorial issues:
- in sec. 4.2, there are missed numbers of equations
- there is no units on included figures, the latency is expresses in sec, time slots? it is impossible do guess.
Author Response
Thanks for your comments! We modified the manuscript and respond your comments as the attached file. Thanks!

Round 2
Reviewer 1 Report
My questions have been well addressed. I do not have further comments.Author Response
Thanks for your comments in improving the manuscript!
Reviewer 2 Report
Thank you for addressing most of my comments. I think the quality of the paper is now improved. Anyway, the most critical issue expressed in my comment 2 has not been addressed at all. "Comment 2: The presented model has not been validated at all. It is strongly required to compare results coming from the model with the simulation results. (...) Please extend evaluation section by including comparison of the results coming from the proposed model and from simulations."
The point is to compare the results calculated by proposed analytical model with the results obtained from simulations. So, please add experiment (or at least include the calculated results on existing figures!) that validate your model, i.e. compare values of the expected latency for a node discovering calculated by the model (eg. 14, 19, etc) with simulation results.
I'm sorry to say but lack of such comparison makes the paper not acceptable...
Author Response
Thanks for the comment and the valuable suggestion! We modified the manuscript according to the comment and we respond the comment as the attached file.
